# Rapfi: Distilling Efficient Neural Network for the Game of Gomoku

## Abstract

Games have played a pivotal role in advancing artificial intelligence, with AI agents using sophisticated techniques to compete. Despite the success of neural network based game AIs, their performance often requires significant computational resources. In this paper, we present *Rapfi*, an efficient Gomoku agent that outperforms CNN-based agents in limited computation environments. *Rapfi* leverages a compact neural network with a pattern-based codebook distilled from CNNs, and an incremental update scheme that minimizes computation when input changes are minor. This new network uses computation that is orders of magnitude less to reach a similar accuracy of much larger neural networks such as Resnet. Thanks to our incremental update scheme, depth-first search methods such as the $\alpha$-$\beta$ search can be significantly accelerated. With a carefully tuned evaluation and search, *Rapfi* reached strength surpassing *Katagomo*, the strongest open-source Gomoku AI based on AlphaZero's algorithm, under limited computational resources where accelerators like GPUs are absent. *Rapfi* ranked first among 520 Gomoku agents on Botzone and won the championship in GomoCup 2024.

## 1 Introduction

Artificial intelligence in board games like Go, Chess, and Shogi has progressed rapidly with the advent of deep neural networks. Notable examples include AlphaGo (Silver et al., 2016), AlphaGo Zero (Silver et al., 2017b), Katago (Wu, 2019), and other efforts in Chess (Schrittwieser et al., 2020) and Shogi (Silver et al., 2017a; Schrittwieser et al., 2020; Nasu, 2018). These methods rely heavily on deep neural networks, requiring powerful accelerators like high-end GPUs. Our objective is to create a specialized network that maintains similar prediction accuracy but runs significantly faster, aiming to outperform current state-of-the-art solutions under constrained computation.

Gomoku, a straightforward yet complex perfect information board game, that serves as an excellent benchmark for evaluating agent performance. Although Gomoku with free openings has been solved (Allis et al., 1994), achieving optimal play with arbitrary balanced openings remains challenging. Current methods often fail to exceed top human players or are computationally intensive. Therefore, we choose Gomoku as our testbed, as it demands precise position evaluation and deep tactical search, making the balance between evaluation and search critical.

Deep models like convolutional neural networks (CNNs) are highly effective at predicting values and policies from 2D inputs, playing a crucial role in reinforcement learning. evaluating them can be computationally intensive. Our key observation is that, in board games with black and white stones, much of this computation is spent extracting features from local patterns. While CNNs excel at this, the process is often redundant, as features of the same patterns remain fixed, and most of the board doesn't change during gameplay. By decomposing the board into local patterns and pre-computing their features, we can greatly reduce computational demands.

To address this, we introduce a lightweight neural network named *Mixnet*, which requires far less computation than CNNs while maintaining comparable accuracy. Our method uses a pattern-based codebook derived from a larger network and includes an incremental update scheme that reduces computation further, especially with depth-first game tree traversal like Alpha-Beta search. We also introduce several improvements in the feed-forward process to enhance the accuracy of value and policy prediction. In summary, our contributions include:

- We decompose a binary board plane into local line-shaped patterns and train a mapping network to convert these patterns into features. This network is then baked into a pattern-indexed codebook for instant lookup, reducing computation by orders of magnitude while maintaining fair evaluation accuracy.

- We introduce an incremental update scheme that minimizes computation when only a few stones on the board change. This mechanism significantly speeds up processing, especially when combined with depth-first search algorithms.

- We integrate several enhancements to optimize the feed-forward computation of value and policy heads, including dynamic policy convolution, value grouping, and star blocks, to improve evaluation accuracy without adding significant computational complexity.

## 2 RELATED WORK

### 2.1 EFFICIENT NEURAL NETWORKS

With the rapid advancement of computer vision, the landscape of neural network design has seen a surge in innovative architectures aimed at minimizing computational demands while maximizing performance. Notable designs include Mixtures of Experts (MoE) (Jacobs et al., 1991), MobileNets (Howard et al., 2017; Sandler et al., 2018; Howard et al., 2019), SENet (Hu et al., 2018), SKNets (Li et al., 2019) and CondConv (Yang et al., 2019). These models are designed to improve both inference speed and prediction accuracy. Unlike the approaches mentioned above, our method takes a novel direction by precomputing a pattern-based feature lookup table and takes advantage of the incremental nature of the gameplay structure. We also incorporate structural improvements in the feedforward components of the network, including depth-wise and point-wise convolutions (Zhang et al., 2020), dynamic convolution (Chen et al., 2020), grouped average pooling, and the star operation (Ma et al., 2024).

### 2.2 SEARCH ALGORITHMS FOR GAME TREES

The most notable game tree search algorithms are Monte Carlo Tree Search (MCTS) (Coulom, 2006) and $\alpha$-$\beta$ search (Knuth & Moore, 1975). MCTS is commonly used by Go agents and has many variants that enhance its performance, such as UCT (Gelly & Wang, 2006), RAVE (Rimmel et al., 2010), and EMCTS (Cazenave, 2007). Additionally, parallel versions of MCTS (Chaslot et al., 2008; Cazenave, 2022) have improved search speed and efficiency on multi-core hardware. On the other hand, $\alpha$-$\beta$ search has a long history with many enhancements, including futility pruning (Heinz, 1998; Hoki & Muramatsu, 2012), razoring (Birmingham & Kent, 1988), null move pruning (Donninger, 1993; Hoki & Muramatsu, 2012), late move reduction (Hoki & Muramatsu, 2012), and history reduction (Schaeffer, 1983). For Gomoku, algorithms like proof-number search (Allis et al., 1994) and threat-space search (Allis et al., 1993) significantly reduce computation for proving positions and improving search efficiency in specific scenarios. Since the traversal order of the game tree may affect the inference speed of our proposed network, we evaluate it using both MCTS and $\alpha$-$\beta$ search in our experiments.

### 2.3 AGENTS FOR THE GAME OF GOMOKU

Gomoku is a two-player game featuring simple rules yet significant strategic depth, making it an excellent testbed for evaluating evaluation models and search techniques. Played on a 15x15 grid, players alternate placing black and white stones, with the black player starting first. The goal is to align five stones in a row—horizontally, vertically, or diagonally. If the board fills without a winner, the game ends in a draw. Research has shown that the first player can always achieve a win, but reaching optimal play from a balanced position remains a complex challenge, with many methods still falling short of optimal performance. Notable advancements in Gomoku agents include threat-space search (Allis et al., 1993), proof-number search (Allis et al., 1993), first-player winning strategies (Allis et al., 1994), adaptive dynamic programming (Zhao et al., 2012), and genetic algorithms (Wang & Huang, 2014). Additionally, several neural network advancements (Wu, 2019; Xie et al., 2018; Wang, 2018) have also been inspired by AlphaZero (Schrittwieser et al., 2020).

## 3 METHODOLOGY

In this section, we present the design of our proposed *Mixnet*. We begin with an overview of the network architecture, followed by the pattern-based feature codebook and an incremental update scheme. Lastly, we describe the feed-forward components, including various improvements to the value and policy prediction heads.

### 3.1 OVERVIEW OF MIXNET ARCHITECTURE

The *Mixnet* takes a one-hot encoded binary board plane $x \in {0, 1}^{2 \times H \times W}$ as input and predicts the policy distribution $\hat{\pi} \in \mathbb{R}^{H \times W}$ and the categorical value $\hat{v} \in \mathbb{R}^3$ (win, loss, and draw rates). Unlike a convolutional neural network, it first decomposes the board into directional line patterns, using these local patterns to construct a feature map. The Mapping network extracts local pattern features, which are then combined by a depth-wise $3 \times 3$ convolution layer. The Mapping network can be exported to a pattern-indexed codebook losslessly, allowing *Mixnet* to efficiently retrieve features during inference with minimal computation. An incremental update scheme further optimizes computation by only recalculating features for updated stones. After building the feature map, lightweight policy and value heads predict the position's evaluation in a feed-forward manner. The pipeline of *Mixnet* is illustrated in Fig. 1.

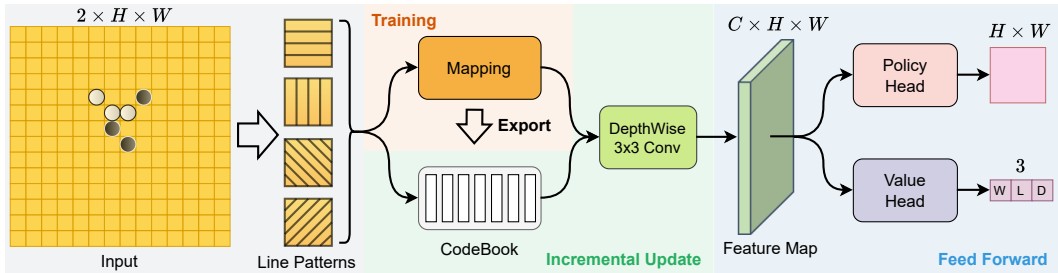

Figure 1: The architecture overview of *MixNet*. It first decomposes a binary board input into local line patterns, then uses a mapping network to generate directional feature maps, which are stored in a pattern-indexed codebook after training. An aggregation and depth-wise $3 \times 3$ convolution are applied with an incremental update mechanism to produce the final feature map. Finally, a policy head and value head predict the policy and win rate in a feed-forward manner.

### 3.2 DECOMPOSING BOARD PLANE AS LOCAL PATTERNS

Consider a $H \times W$ sized board with each cell can be Black, White, or Empty, represented as $S = \{0, 1, 2\}^{H \times W}$ (with 0 for Empty, 1 for Black, and 2 for White). As the enormous state space would be too large to store, we break it down into local patterns with manageable state sizes. Since line connections are crucial in Gomoku, we organize these local patterns into line segments in various directions, as illustrated on the left side of Fig. 2.

For each point $(i, j)$ in the $i$-th row and $j$-th column, we define four local line patterns: the horizontal pattern $L_{i,j}^{(0,1)}$, the vertical pattern $L_{i,j}^{(1,0)}$, the main diagonal pattern $L_{i,j}^{(1,1)}$, and the anti-diagonal pattern $L_{i,j}^{(1,-1)}$. Each line segment has a length of $11$, enabling it to capture the connection features of the surrounding five stones. Thus, the localized line pattern at this point can be represented as:

$$L_{i,j}^{(m,n)} = \{s(i + m * k, j + n * k)| -5 \le k \le 5\}. \tag{1}$$

To construct a feature map $F \in \mathbb{R}^{C \times H \times W}$ from the board state $S$, we define a mapping function $M : L_{i,j}^{(m,n)} \rightarrow f_{i,j}^{(m,n)} \in \mathbb{R}^C$ that transforms a line pattern at point $(i, j)$ into the corresponding feature with $C$ channels. We use two mapping functions: $M_{hv}$ for the horizontal and vertical patterns, and $M_{di}$ for the main diagonal and anti diagonal patterns. The final feature at point $(i, j)$, denoted as

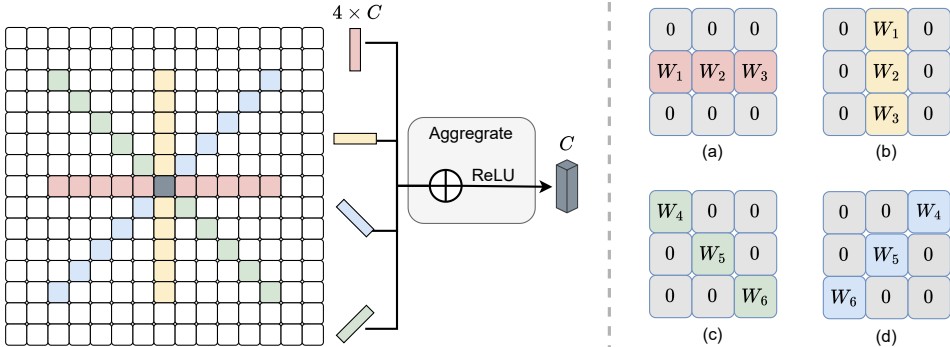

Figure 2: Left: Line patterns in four directions o a $15 \times 15$ board. By applying the mapping functions $M_{hv}$ and $M_{di}$ on these patterns, we obtain four features at this point, which are aggregated by summing and applying a ReLU activation to produce the final feature. Right: Convolution kernels for horizontal, vertical, main diagonal, and anti-diagonal patterns.

$f_{i,j} \in \mathbb{R}^C$, is obtained by aggregating the features from the four directions, and then applying a ReLU activation to introduce non-linearity:

$$f_{i,j} = \text{ReLU}(M_{hv}(L_{i,j}^{(0,1)}) + M_{hv}(L_{i,j}^{(1,0)}) + M_{di}(L_{i,j}^{(1,1)}) + M_{di}(L_{i,j}^{(1,-1)})). \tag{2}$$

Finally, we obtain the feature map $F$ by concatenating features at each point:

$$F = \bigcup_{i=1}^{15} \bigcup_{j=1}^{15} f_{i,j} \tag{3}$$

### 3.3 Distilling Pattern-based Codebook

Considering the border, there are $N = \sum_{i=0}^{5} \sum_{j=0}^{5} 3^{i+1+j} = 397488$ possible line patterns. To learn features of these patterns, one might use an embedding layer of size $N$ to retrieve features via a pattern's index. However, this approach is prone to overfitting and may not sufficiently train all features, as not every pattern is guaranteed to appear often enough to generate meaningful gradients. Instead, we utilize a specialized Mapping network with a $3 \times 3$ kernel that performs convolution operations along specific directional lines. The kernels, shown in Fig. 2, have non-zero weights only in the designated direction. Different weights are used for horizontal/vertical and diagonal directions to capture subtle differences caused by directionality. We refer to this convolution layer as *Dir Conv*.

The structure of the mapping networks is depicted on the left side of Fig. 3. With a $3 \times 3$ reception field per layer, we use five *Dir Conv* layers to ensure the network captures line segment patterns of length 11. Additionally, we incorporate point-wise $1 \times 1$ convolution layers alternately to enhance feature extraction and apply skip connections to facilitate smoother training. The mapping network operates on an internal feature map with $M$ channels and uses the final point-wise convolution to produce the output directional feature map with $C$ channels.

Since the mapping network functions as a convolution network with shared parameters, it can be trained robustly with a limited amount of data. Once training is complete, we pre-compute the mapping network by enumerating all $N$ patterns as a plane of shape $2 \times 1 \times (i+1+j)$ and feed them into the network. We rearrange all kernel weights of *Dir Conv* as shown in Fig. 2 (a). By recording the feature outputs at $(1, i+1)$, we export the mapping network losslessly as a pattern-indexed codebook $f_{CB} \in \mathbb{R}^{N \times C}$, which contains features of all possible patterns.

### 3.4 Incremental Update of Feature Maps

After the mapping phase, we convert the patterns into four directional feature maps, which we combine into a single feature map $F \in \mathbb{R}^{C \times H \times W}$ using the aggregation operation in Eq. 2. To further enhance the receptive field, we apply a depth-wise $3 \times 3$ convolution to the first half of the

Figure 3: Left: The mapping network takes a board plane and outputs four directional pattern feature maps. Right: The incremental updateable parts of *Mixnet*, which includes aggregation of four directional features and a depth-wise $3 \times 3$ convolution layer that operators on half of the channels.

channels. The processed feature map $F' \in \mathbb{R}^{C \times H \times W}$ at point $(i, j)$ of the $k$-th channel is given as:

$$F'(k, i, j) = \sum_{m=1}^{3} \sum_{n=1}^{3} F(k, i + m - 2, j + n - 2) \cdot W(c, m, n), \ k \in [1, C/2] \tag{4}$$

where $W \in \mathbb{R}^{(C/2) \times 3 \times 3}$ represents the convolution weights, and $F(c, i, j)$ uses zero padding for out-of-bounds elements. The last half of the channels retain the original feature map to minimize computation: $F'(k, i, j) = F(k, i, j), \ k \in [C/2, C]$.

Since the features arise from localized patterns, we propose an approach called incremental update of feature maps to further reduce computational cost. Instead of recalculating features for the entire board after a move, we only update those for affected positions. Changes to one or a few stones require updating only necessitates updating the affected features from the directional feature maps by looking up the codebook with the new pattern's index. For instance, if a single stone changes, at most $4 \times 11$ directional features are affected, as illustrated on the left side of Fig. 2, allowing us to recompute only these features to update $F$. Similarly, we only update the impacted regions in the processed feature map $F'$. By maintaining an accumulator for $F'$ and adding the delta activation values, we significantly reduce the computation needed to obtain the latest $F'$ after the depth-wise convolution. Experiments in Sec. 5.2 demonstrate that this optimization's speed advantage is particularly significant when combined with a depth-first game tree search.

## 3.5 ENHANCING FEED-FORWARD HEADS

After the incremental update phase, we compute the policy and value heads from the processed feature map $F'$ in a feed-forward manner. To improve the accuracy of *Mixnet* without a significant increase in computation, we propose three enhancements: dynamic policy convolution, value grouping, and star block.

Dynamic convolution enhances policy prediction accuracy, as illustrated in Fig. 4 (a). The policy head first applies average pooling on $F'$ to compute the global feature mean, followed by two linear layers with a ReLU activation to generate the weights and bias for dynamic point-wise convolution, which is applied to the first $P$ channels of $F'$. A subsequent point-wise convolution transforms the 16-channel policy features into the raw policy output $\hat{\pi} \in \mathbb{R}^{H \times W}$. As shown in Sec. 3, this dynamic convolution enables the policy head to capture global board information and adaptively adjust predictions by modulating channel contributions, significantly improving accuracy with minimal computational cost.

For the value head, shown in Fig. 4 (b), we implement two methods to enhance prediction accuracy: value grouping and the star block. The value grouping module divides the input feature map $F'$ into $3 \times 3$ regions, creating local feature chunks denoted as $F'^{chunk}_{i,j} \in \mathbb{R}^{C \times H' \times W'}$, where $i, j \in \{1, 2, 3\}$.

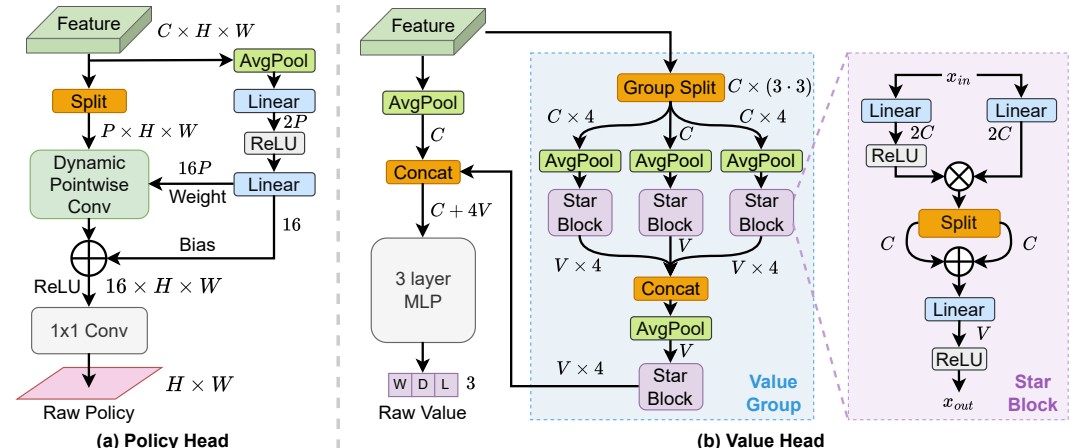

Figure 4: Structure of the policy head and the value head.

This includes four corner chunks, four edge chunks, and one center chunk. We apply average pooling to each chunk to obtain the mean feature, and then use the star block to transform each mean feature into value group features $G_{i,j} \in \mathbb{R}^V$. These features are concatenated and averaged to form $2 \times 2$ groups of intermediate features $G'_{i,j} \in \mathbb{R}^V$, where $i, j \in \{1, 2\}$:

$$G'_{i,j} = (G_{i,j} + G_{i,j+1} + G_{i+1,j} + G_{i+1,j+1})/4 \tag{5}$$

Next, we concatenate the global feature mean with these four group features $G'_{i,j}$ after applying another star block, and use a three-layer MLP to produce the final raw value estimation. The star block, illustrated on the right in Fig. 4 (b), takes the input tensor through two separate linear layers (one with a ReLU activation) to double the channels, followed by multiplication between them. A pairwise dot product is then applied to halve the channels, with a final linear layer and ReLU activation to produce the output. This multiplication pooling acts similarly to the kernel trick(Ma et al., 2024), enhancing the non-linearity of this relatively shallow network. Overall, the introduction of value grouping and the star block improves the value head's prediction accuracy by effectively integrating local and global features while keeping computational demands minimal.

## 4 EXPERIMENTAL SETUP

### 4.1 DATASET

To evaluate our proposed model's performance, we first train all models in a supervised manner and then conduct strength tests under various configurations. The dataset, generated by *Katagomo* (Hang, 2024) through an AlphaZero-like self-play process over several weeks, contains approximately 30.8 million positions. Each position is represented as a 3-tuple $(B, V_t, \pi_t)$. $B \in \{0, 1\}^{2 \times H \times W}$ is the board input, with channels representing the current player's and opponent's stones.. $V_t = (p_w, p_l, p_d) \in \mathbb{R}^3$ is the value target, indicating win, loss, and draw probabilities.. $\pi_t \in \mathbb{R}^{H \times W}$ is the normalized policy target.

### 4.2 TRAINING DETAILS

To assess the tradeoff between accuracy and speed, we assess three sizes of *MixNet*. The smallest configuration has $M = 64, C = 32$ for mapping, and $P = 16, V = 32$ for policy and value heads, with each larger model doubling the channel sizes. We present the parameter counts, computational cost, storage size, and the average inference speed for a specific search algorithm in Tab. 1.

To evaluate strength and efficiency, we use the widely used ResNet architecture (He et al., 2016), which includes two skip-connected convolution layers with $f$ channels in each of $b$ blocks. A $3 \times 3$ convolution kernel processes the input, followed by a $1 \times 1$ kernel, global pooling, and an MLP head for the final value and policy outputs. Details on the baselines can be found in the appendix.

| Model | k#Params | | kFLOPs | Storage | Infer. Speed | |
|---|---|---|---|---|---|---|
| | CB | FF | (forward) | (in MiB) | MCTS | $\alpha$-$\beta$ |
| Mixnet small | 14160 | 37 | 51 | 28.4 | 47575 | 428K |
| Mixnet Medium | 28320 | 146 | 131 | 54.7 | 36823 | 257K |
| Mixnet Large | 56640 | 580 | 381 | 111 | 18401 | 104K |
| Resnet 4b64f | 0 | 298 | 67089 | 1.13 | 1064 | 1758 |
| Resnet 6b96f | 0 | 1001 | 225396 | 3.81 | 297 | 484 |
| Resnet 10b128f | 0 | 2960 | 666403 | 11.2 | 95 | 163 |
| Resnet 15b192f | 0 | 9976 | 2245493 | 38.0 | 28 | 51 |
| Resnet 20b256f | 0 | 23631 | 5318727 | 90.0 | 11 | 32 |

Table 1: The number of CodeBook and Feed-Forward parameters, theoretical inference FLOPs, weight size and average inference speed of different models. Inference speed is measured using MCTS (playouts per second) and $\alpha$-$\beta$ search (nodes per second) respectively.

All models are trained using the Adam optimizer with an initial learning rate of $lr = 0.001$, $\beta_1 = 0.9$, $\beta_2 = 0.999$ and $\epsilon = 10^{-8}$, with a batch size of 128 samples for 600k iterations. For training *MixNet*, we use cross-entropy loss and knowledge distillation with a Resnet-6b128f pretrained on the same dataset as the teacher. Further training details are available in the appendix.

### 4.3 SEARCH

A key difference between algorithms for general reinforcement learning and perfect-information board games is that the latter can use "look-ahead" search to reduce approximation errors from imperfect evaluations. Improving evaluation and search often conflict due to their computational demands, creating a tradeoff. Our goal is to find the optimal balance that maximizes performance within a fixed computation budget. To assess model effectiveness, we need to test them with a practical search algorithm, considering the variability in evaluation accuracy and inference speed.

We validate the efficiency of our proposed model using two search algorithms. First, we employ the best-first Monte Carlo Tree Search (MCTS) with the Predictor Upper Confidence Bounds applied to Tree (PUCT) variant, which combines policy and value for selective search. Given that *MixNet* benefits from incremental updates suited for depth-first traversal, we also test it with Alpha-Beta search, implementing the Principal Variation Search (PVS) variant and incorporate various enhancements that improve performance. Further details on our implementations of these search algorithms are available in the appendix.

## 5 RESULTS

We begin by comparing the loss and accuracy of various models during supervised training. Next, we evaluate the relative strength of these models combined with search algorithms given a fixed computation budget. Finally, we conduct an ablation study.

### 5.1 LOSS AND ACCURACY COMPARISON

We present the supervised training loss and relative ELO for all models in Tab. 2, based on a fixed amount of search. The train and validation losses indicate how effectively each model learns static evaluation for both value and policy prediction. Relative ELO is determined by running 400 games with varying search nodes or playouts. From the table, it's clear that our proposed *MixNet* excels in learning value but struggles more with policy learning. In contrast, ResNet shows significant policy improvements with additional residual blocks. *MixNet*'s value loss is comparable to ResNet-6b94f, while its policy loss only approaches that of ResNet-4b64f at its largest configuration. We believe this is due to the stacked convolution layers offering a larger receptive field, which is crucial for effective policy prediction.

This is also evident in the performance of raw neural networks; using MCTS with a single playout effectively relies solely on the network's policy prediction. Here, all *MixNet* configurations

| Model | Supervised Loss | | | | Relative ELO Under Fixed Nodes | | | | | | | |
| | Train | | Validation | | MCTS | | | | $\alpha$-$\beta$ Search | | | |
| | Value | Policy | Value | Policy | 1p | 4p | 16p | 64p | 1d | 3d | 5d | 7d |
|---|---|---|---|---|---|---|---|---|---|---|---|---|
| Mixnet Small | 0.7925 | 1.278 | 0.7836 | 1.277 | 0 | 0 | 0 | 0 | 0 | 0 | 0 | 0 |
| Mixnet Medium | 0.7791 | 1.213 | 0.7715 | 1.214 | 64 | 64 | 68 | 90 | 50 | 54 | 49 | 82 |
| Mixnet Large | 0.7688 | 1.165 | 0.7632 | 1.166 | 101 | 121 | 119 | 138 | 77 | 84 | 89 | 102 |
| Resnet 4b64f | 0.8004 | 1.188 | 0.7928 | 1.184 | 106 | 97 | 64 | 37 | 31 | 30 | 25 | 48 |
| Resnet 6b96f | 0.7727 | 1.035 | 0.7645 | 1.031 | 247 | 212 | 208 | 214 | 123 | 154 | 129 | 155 |
| Resnet 10b128f | 0.7534 | 0.9246 | 0.744 | 0.9215 | 354 | 320 | 336 | 361 | 198 | 251 | 247 | 270 |
| Resnet 15b192f | 0.7329 | 0.8376 | 0.7303 | 0.8384 | 475 | 391 | 442 | 505 | 228 | 320 | 338 | 352 |
| Resnet 20b256f | 0.7264 | 0.7979 | 0.7181 | 0.7969 | 516 | 451 | 507 | 566 | 283 | 370 | 404 | 415 |

Table 2: Comparison of different models' value and policy losses and their relative strength under a fixed amount of search nodes.

performed worse than ResNets, which have superior policy predictions. However, as more fixed nodes are introduced, the search increasingly utilizes value prediction to identify the best root move. Consequently, *MixNet*'s strength gradually surpasses that of the smallest ResNet-4b64f, while the performance gap with ResNet 6b96f-narrows.

Thus, in terms of fitting accuracy, *MixNet* aligns roughly between ResNet 4b64f and ResNet 6b96f. However, it reaches this level of accuracy with significantly lower inference computation, enabling much faster evaluations. This speed can be a critical advantage when operating under fixed computation budgets rather than a fixed number of search nodes.

## 5.2 STRENGTH COMPARISON WITH SEARCH

Our proposed *MixNet* prioritizes efficiency with a well-distilled pattern-based codebook and enhanced feed-forward head, leading to significantly reduced inference computation compared to a ResNet baseline. As shown in the last two columns of Tab. 1, *MixNet*'s inference throughput is orders of magnitude higher than that of ResNet. Furthermore, when paired with a depth-first $\alpha$-$\beta$ search instead of best-first MCTS, *MixNet*'s evaluation computation decreases even further, leveraging the incremental update mechanism between closely searched positions. Although *MixNet* may not surpass larger neural networks in raw value and policy evaluation accuracy, its much faster inference throughput offers a significant advantage when used with search algorithms.

We evaluate the efficiency of *MixNet* against baselines using the vanilla Monte Carlo Tree Search from Sec. 4.3, which relies heavily on the models' value and policy predictions. We conduct 400 time-controlled games between each pair of models across various time settings to assess their relative strengths. To ensure accurate and fair measurements, games begin from balanced openings sampled from a prepared book, with each model playing as the first player once per opening.

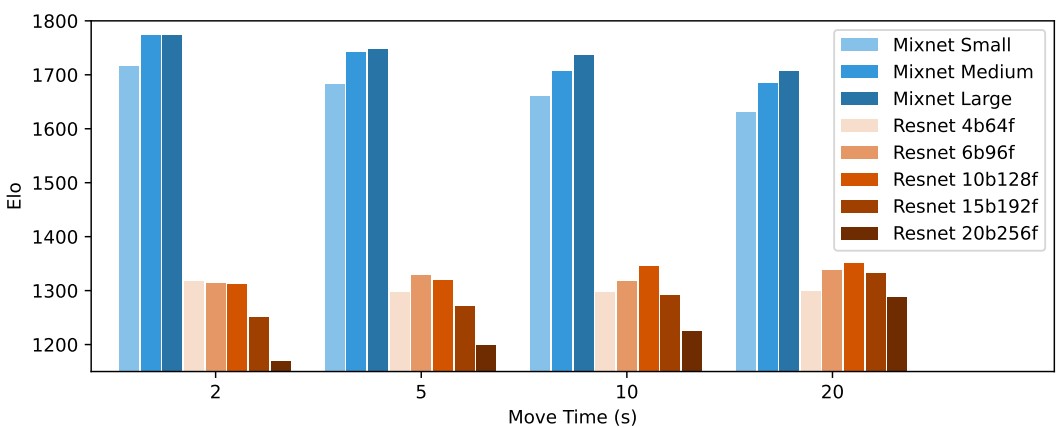

Figure 5: ELO of all models given a fixed move time of Monte Carlo Tree Search.

As shown in Fig. 5, all *MixNet* configurations consistently outperform ResNet baselines by a significant margin of 300-400 ELOs, translating to a winning rate of $85\%$-$92\%$. Notably, larger *MixNet* models, despite slower inference speeds, achieve higher strength, especially with more time allocated for searching. This trend is also seen in ResNet models, where stronger performance emerges with larger models and increased move time, and the ELO gap with *MixNet* narrows slightly. It indicates that with more search time and established tactics, more accurate evaluations gain importance. Nonetheless, even with 20 seconds per move, *MixNet*'s advantages remain clear. Since both sides use the same move time, these results reflect the true strength of the models within a constrained computational budget in real-world scenarios.

## 5.3 Strength Comparison between Agents

To explore the limits of *MixNet*, we combine it with the advanced Alpha-Beta search described in Sec. 4.3. We also include the state-of-the-art Gomoku agent *Katagomo* (Hang, 2024) for strength comparison. This version of *Katagomo* utilizes domain-specific search and several model enhancements to improve performance while lowering the computational cost of large convolution neural networks, having undergone months of reinforcement learning. Since *Katagomo* is primarily designed for GPUs, we use its CPU version to ensure a fair comparison within the same computation resource constraints.

As shown in Fig. 6, $\alpha$-$\beta$ search exhibits significantly higher playing strength due to its depth-first traversal of the game tree, effectively leveraging the speed of our incremental update mechanism. Notably, the large *MixNet* does not outperform the medium and small alternatives, likely due to the higher codebook update cost diminishing the benefits of incremental updates. Nevertheless, combining *MixNet* with $\alpha$-$\beta$ search demonstrates the speed enhancement of incremental updates, achieving a strength of approximately 400 ELO above the SOTA agent in a CPU-only environment with limited computational resources.

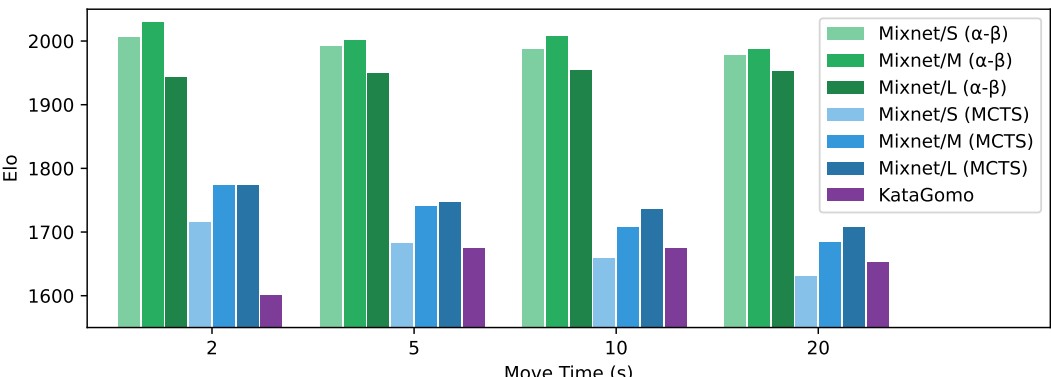

Figure 6: ELO of various *MixNet* agents and *Katagomo* given a fixed move time of search.

## 5.4 Ablation Studies

To validate the effectiveness of enhancements in *MixNet*'s feed-forward components, we perform an ablation study by removing specific modules and comparing their loss and ELOs under identical computation time and search algorithms in Tab. 3. Specifically, We evaluate the impact of the star block, value group, and dynamic convolution. The star block is replaced with a single linear layer, the value grouping is simplified by retaining only the global feature mean for the value MLP, and the dynamic policy convolution is substituted with a fixed point-wise convolution of the same size.

Training losses show a significant increase in value loss when either the star block or value group is removed, suggesting that both components enhance the capture of localized features, which aid in value prediction alongside the global feature mean. Conversely, the policy loss rises considerably when the dynamic policy convolution is eliminated, indicating that the global information from the global feature mean is crucial for the policy convolution, which relies solely on localized features to make location-specific decisions.

| Mixnet Configuration | Train Loss | | MCTS | | | | $\alpha$-$\beta$ Search | | | |
|---|---|---|---|---|---|---|---|---|---|---|
| | Value | Policy | 2s | 5s | 10s | 20s | 2s | 5s | 10s | 20s |
| Small (full) | 0.7925 | 1.278 | 0 | 0 | 0 | 0 | 0 | 0 | 0 | 0 |
| Small w/o star block | 0.7995 | 1.294 | -14 | -15 | -30 | -4 | 5 | -6 | -1 | -7 |
| Small w/o value group | 0.8083 | 1.287 | -72 | -82 | -84 | -82 | -70 | -77 | -70 | -81 |
| Small w/o dynamic conv | 0.7903 | 1.465 | -143 | -144 | -150 | -145 | 2 | -1 | 3 | -8 |
| Medium (full) | 0.7791 | 1.213 | 54 | 60 | 50 | 62 | 11 | 10 | 20 | 9 |
| Medium w/o star block | 0.7839 | 1.223 | 26 | 29 | 19 | 30 | 2 | -4 | 5 | -2 |
| Medium w/o value group | 0.7957 | 1.219 | 2 | 2 | -6 | 14 | -43 | -47 | -42 | -61 |
| Medium w/o dynamic conv | 0.7801 | 1.431 | -118 | -116 | -120 | -113 | -24 | -32 | -12 | -21 |
| Large (full) | 0.7688 | 1.165 | 57 | 74 | 77 | 84 | -82 | -58 | -45 | -38 |
| Large w/o star block | 0.7733 | 1.160 | 90 | 99 | 96 | 112 | -9 | -11 | -3 | -8 |
| Large w/o value group | 0.7844 | 1.161 | 28 | 41 | 33 | 48 | -24 | -34 | -42 | -29 |
| Large w/o dynamic conv | 0.7712 | 1.387 | -112 | -97 | -99 | -88 | -100 | -92 | -66 | -61 |

Table 3: *Mixnet* Elo ratings for different models and move times.

Tab. 3 shows the relative ELOs when removing these modules. For MCTS, policy prediction accuracy is critical; removing dynamic policy convolution results in over a 100 ELO loss. Notably, MCTS depends less on value accuracy when a strong policy is in place, resulting in the larger *MixNet* without the star block achieving the best performance, likely due to the star block's significant speed reduction. In contrast, for $\alpha$-$\beta$ search, a medium-sized model is preferred, with value accuracy playing an essential role, making the full medium *MixNet* the top performer in terms of ELOs.

## 6    CONCLUSION

We present *Rapfi*, an efficient Gomoku AI agent optimized for resource-limited environments. Our proposed model, *Mixnet*, decomposes the board plane into local line-shaped patterns and distills a pattern-indexed codebook from a specially designed mapping network. This approach, combined with an incremental update scheme and enhancements in the feed-forward heads, allows *Mixnet* to match the performance of large CNNs while drastically reducing computational requirements. Experimental results show that *Mixnet* significantly outperforms Resnet baselines and the state-of-the-art Gomoku agent *Katagomo* under the same computational constraints. Combining *Mixnet* with a carefully tuned Alpha-Beta search, *Rapfi* ranks top among 520 Gomoku agents on Botzone, and successfully won the championship in the 2024 Gomocup tournament against 54 competitors. Details of the match results are available on the website of Botzone and Gomocup.

In conclusion, we have showcased the potential of carefully designed compact neural networks, especially in scenarios where evaluation speed is crucial for agent performance. The success of our model stems from the novel pattern decomposition and the precomputation of a pattern-indexed feature codebook. Our incremental update mechanism further minimizes computational costs when inputs change partially, making it well-suited for game tree traversal with sequential evaluations of similar positions. Additionally, enhancements in feed-forward heads significantly boost prediction accuracy without adding excessive computation costs. However, this paper does not address certain limitations, such as the model's shallowness and its scalability to larger networks. We hope our findings will inspire further research into efficient neural networks and advancements in game AI.

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

# A APPENDIX

## A.1 DETAILS OF RESNET BASELINE

The detailed architecture of the ResNet baselines used in our experiments is illustrated in Fig. 7. The model takes an input board tensor with a shape of $2 \times H \times W$. Initially, the input passes through a $3 \times 3$ convolutional layer with $2 \times f$ filters, resulting in an output shape of $f \times H \times W$, followed by a ReLU activation function. This output then enters a module containing residual blocks, repeated $b$ times. Each residual block consists of two $3 \times 3$ convolutional layers with $f \times f$ filters, each followed by a ReLU activation function, and a residual connection that adds the block's input to the output of the second convolutional layer.

The output of the residual block module is processed through two parallel paths for feature aggregation: the first path includes an average pooling layer, a linear layer, and a ReLU activation function, yielding an output with dimensions 3; the second path consists of a $1 \times 1$ convolutional layer with $f$ filters and a ReLU activation function, producing an output with dimensions $H \times W$. This ResNet architecture effectively captures spatial features through convolutional layers and residual connections, while the parallel paths ensure comprehensive feature aggregation and transformation. We implement the ResNet baselines using the ONNX runtime, which includes various optimizations for CPU inference.

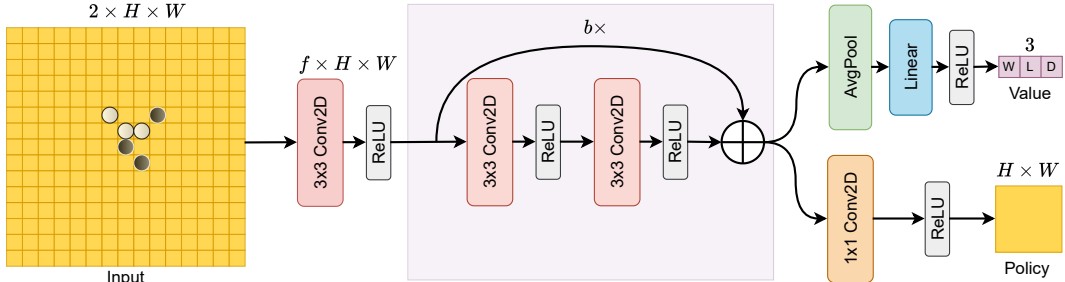

Figure 7: The structure of Resnet baselines used in our experiments.

## A.2 DETAILS OF TRAINING

Our model is trained end-to-end, with both the policy and value heads optimized using *policy* and *value* objectives. In this section, we provide a detailed description of the loss functions related to these targets:

- **Policy Target**

  The Policy Target objectives are derived from the self-play data generated by KataGo.

  $$\mathcal{L}_p = - \sum_{m \in \text{ moves}} \pi(m) \log(\hat{\pi}(m)) \tag{6}$$

  where $\pi$ is the policy target and $\hat{\pi}$ is the model's prediction of $\pi$ and moves $\in \{1, 2, ..., H \times W\}$.

- **Value Target**

  The Value Target objectives are also derived from the self-play data generated by KataGo.

  $$\mathcal{L}_v = - \sum_{n \in \{1,2,3\}} V_t(n) \log(\hat{V}(n)) \tag{7}$$

  where $V_t$ is the value target from the game result and $\hat{V}$ is the model's value prediction, both have 3 channels representing the probability of win, loss, and draw.

The final loss function is the sum of $loss_p$, $loss_v$, and a regularization term.

$$\mathcal{L} = \mathcal{L}_p + \mathcal{L}_v + \lambda \sum_{i=1}^{n} w_i^2 \tag{8}$$

where $w_i$ is parameter of the model.

In training *Mixnet*, we utilize knowledge distillation with a teacher model based on the ResNet baseline from Sec. A.1, specifically configured as ResNet 6b128f ($b = 6$, $f = 128$). Initially, we train ResNet 6b128f to convergence to serve as the teacher model, which then aids in training *MixNet* by providing enriched data that helps lower the loss. The outputs of ResNet 6b128f are used as supervisory labels for *MixNet*. For loss calculation, we adopt a mixed loss strategy, incorporating 75% from the distillation labels and 25% from the true labels.

### A.3 DETAILS OF MIXNET IMPLEMENTATION

When implementing the inference of *Mixnet*, we quantize the whole neural network to avoid any floating point error that may be introduced during the accumulation process of incremental update, and further speed up the inference computation. Specifically, we clamp all features in the pattern-based codebook to $[-16, 16]$, and quantize them into 16-bit integer using a scale factor of 32. For the incremental update computation, we fetch the directional features and computes both the aggregation operator and the depth-wise $3 \times 3$ convolution in 16-bit integer, then the sum of final feature map is accumulated in 32-bit integer. For the policy head, we also use 16-bit quantized matrix multiply for the dynamic convolution, and revert to floating point for the final point-wise convolution. For the value head and the linear layers that produces the dynamic weights and bias in the policy head, we use 8-bit matrix multiply with 32-bit accumulation quantization for all linear layers.

To completely utilize the computation power of modern CPUs, we implement the inference of *Mixnet* with Single Instruction Multiple Data (SIMD) intrinsics. Specifically, we use the AVX2 instruction set on the Intel CPU platform to speed up the processing of incremental update as well as the feed-forward heads. This optimization achieves about roughly 4x inference speed compared to a non-vectorized implementation.

### A.4 DETAILS OF SEARCH ALGORITHMS

In this section, we detail the implementation of two search algorithms used in our experiments: Monte Carlo Tree Search (MCTS) and Alpha-Beta Search.

MCTS is a selective best-first search algorithm that iteratively expands a search tree through four steps: selection, expansion, evaluation, and backpropagation. It recursively selects the child node with the highest upper confidence bound until reaching an unexpanded leaf node. The neural network model is then evaluated to obtain the value and policy of that position, followed by backpropagating the results to update the average utility and visit count of all ancestor nodes. We employ a slightly modified version of the Predictor Upper Confidence Bounds applied on Trees (PUCT) variant in our experiments. Specifically, at each state $s$, for every time step $t$, a best action $a_t$ is chosen using the following selection formula:

$$a_t = \arg\max_a (Q(s_t, a) + U(s_t, a)), \tag{9}$$

where

$$U(s_t, a) = c_{\text{puct}}(s_t) \frac{\sqrt{\sum_b N(s_t, b)}}{1 + N(s_t, a)}, \tag{10}$$

and $Q(s_t, a)$ and $N(s_t, a)$ is the average utility and the visit count of the child $a$. We use a dynamic PUCT factor that scales with the visit count of the parent node:

$$c_{\text{puct}}(s_t) = c_{\text{puct-init}} + c_{\text{puct-log}} \cdot \log(1 + \frac{\sum_b N(s_t, b)}{c_{\text{puct-base}}}), \tag{11}$$

where $c_{\text{puct-init}} = 1.0$, $c_{\text{puct-log}} = 0.4$, $c_{\text{puct-base}} = 500$. For unexpanded child nodes that does not have an average utility value, we use the first-play urgency heuristic:

$$Q(s_t, a) = Q(s_t) - c_{\text{fpu}} * \sqrt{\sum_b P(s_t, b)\mathbb{I}(N(s_t, b) > 0)}, \tag{12}$$

where $c_{\text{fpu}} = 0.1$, $Q(s_t)$ is the average utility value of the parent node, and $\sum_b P(s_t, b)\mathbb{I}(N(s_t, b) > 0)$ represents the sum of policy of all explored children. Additionally, we implement several enhancements to MCTS, including graph search, which treats the search tree as a directed acyclic graph to minimize redundant node computations. We also apply Lower Confidence Bounds at the root node by tracking the utility variance at each node and select the best move that maximizes the lower bound. Finally, delayed policy evaluation conserves memory by only generating and evaluating child nodes upon a second visit to a node.

Alpha-Beta ($\alpha$-$\beta$) search is a depth-first algorithm that exhaustively searches up to a specified depth while tracking upper and lower bounds to prune irrelevant branches. Its efficiency hinges on move ordering; perfect ordering can reduce complexity from exponential to square root of exponential growth. We utilize the Principle Variation Search (PVS) variant, which employs zero-window searches for non-PV moves to further enhance pruning opportunities. However, even with perfect ordering, the algorithm's time complexity grows exponentially with depth due to its exhaustive nature, and it is susceptible to the horizon effect. To strengthen $\alpha$-$\beta$ search, we integrate quiescence search commonly used in chess engines, focusing on moves for the attacker side that allow only one response from the defender, thus reducing the branch factor and uncovering tactical paths hard for pure evaluation models. A transposition table stores search results to optimize first moves, and we apply various enhancements like futility pruning, late move reduction, singular extension and null move pruning. Additionally, we leverage neural network policy predictions to rank moves and adjust search depth dynamically. Overall, these enhancements, combined with parameter tuning via the Simultaneous Perturbation Stochastic Approximation (SPSA) algorithm, result in significant ELO improvements over the basic $\alpha$-$\beta$ search.

