# OpenReview forum: "Rapfi: Distilling Efficient Neural Network for the Game of Gomoku"
_ICLR.cc/2025/Conference — ICLR 2025 Conference Withdrawn Submission_

### Official Review · Reviewer_rmMJ · 2024-10-30

**Soundness:** 2
**Presentation:** 3
**Contribution:** 2
**Rating:** 3
**Confidence:** 4

**Summary:**

This paper presents Rapfi, a Gomoku agent exhibiting strong performance under limited computational constraints. Central to Rapfi is an architecture named MixNet tailored to the game of Gomoku. MixNet uses directional convolutional layers to extract local line patterns. The authors also pre-compute a codebook of all the line patterns for fast access to the features. In addition, they propose an incremental update scheme to reduce computation overhead when the change is local. They further propose dynamic policy convolution, value group and star block to enhance the feed-forward heads. The empirical study demonstrates that MixNet matches the absolute strength of a medium-sized residual neural network(ResNet) counterpart but outperforms ResNet of all sizes with a large margin under the same computational budget because Rapfi can perform orders of magnitude more Monte Carlo tree search (MCTS) than the ResNet agent within the same amount of time.

**Strengths:**

- The writing of the paper is clear, with detailed figures to illustrate the MixNet architecture.
- The MixNet is an original idea that, along with the proposed enhancements, contributes to Rapfi's computational efficiency.
- The evaluation is rigorous, with abundant runs for each trial. The ablation study is also thorough.
- Rapfi has significantly higher playing strength against the ResNet counterpart under strict computation constraints.

**Weaknesses:**

Major concern:
- The design of the line patterns and the codebook heavily exploits Gomoku's structure, making the technique hardly applicable to other tasks. For more complex games, such as go and chess, the patterns are challenging to specify or enumerate exhaustively. Thus, it is hopeless to build a codebook that covers every possible pattern. The architecture of MixNet, including the directional convolutions, is also tailored and engineered exclusively for the game of Gomoku. In addition, human-crafted patterns from domain knowledge are unlikely to scale [1]. I understand the authors are trading the scalability for speed, and I do not intend to diminish Rapfi's achievement in the Gomoku community. Nevertheless, as far as the focus of ICLR is concerned, I believe the contribution is narrow and limited.
- Table 2 and Figure 5 reveal that MixNet outperforms the ResNets only in the scenario of limited search. With more search iterations, the performance of the ResNet agents quickly catches up, eventually surpassing MixNet. This further testifies that MixNet does not improve as fast as ResNet with deeper searches.
- The ablation study shows that the star block detriments the performance of the larger MixNet, making it a questionable technique.

Minor issues:
- > Research has shown that the first player can always achieve a win, ...

This claim requires a citation.
- > Additionally, several neural network advancements (Wu, 2019; Xie et al., 2018; Wang, 2018) have also been inspired by AlphaZero.

The citation for AlphaZero is MuZero.
- > which we combine into a single feature map F ∈ RC×H×W using the aggregation operation in Eq. 2.

Wrong equation number. It should be 3 instead.

[1] Sutton, R. (2019). The bitter lesson. Incomplete Ideas (blog), 13(1), 38.

**Questions:**

Regarding Figure 5, are the Elo ratings relative ratings within each group of the move time instance? If they are, I recommend using a common baseline for all the groups. The current figure gave me the first impression that the strength of MixNets decreases with more time to search, which is pretty counterintuitive and causes some confusion.

---

### Official Review · Reviewer_EwUk · 2024-11-02

**Soundness:** 2
**Presentation:** 2
**Contribution:** 2
**Rating:** 3
**Confidence:** 3

**Summary:**

This work centers around a newly proposed network architecture, Mixnet, that when used in conjunction with another learning algorithm, won a public online tournament for the game of Gomoku.

The benefits of using Mixnet compared to contemporary network architectures is its inference speed; Mixnet boasts inference speed several magnitudes higher than ResNet networks.  (Table 1)

To achieve this speed, Mixnet leverages several components that specifically benefit its application in the game of Gomoku.

First, it decomposes the board space as local line patterns.
Second, Local line patterns are converted into features using an additional mapping network.
Third, the mapping network is then distilled into a codebook for instant look up.
fourth, introduce an incremental update scheme to further reduce computation.

They train Mixnet in a supervised fashion using policy and value estimates learned by Katagomo.

To assess the effectiveness of their network architecture. they pair Mixnet with two search algorithms, MCTS and Alpha-Beta search.

They then compare their network with ResNet (ResNet is also paired with the same underlying search algorithms) and Katagomo, the strongest open-source Gomoku agent based on AlphaZero.

Their comparative measure uses an ELO rating obtained after a fixed number of games (400).
The ELO tournament is repeated using different amounts of fixed move time (2s, 5s, 10s, 20s)

They demonstrate that their agents (depending on either MCTS or Alpha-Beta underlying algorithm) score a higher ELO after a certain number of matches compared to ResNet and Katagomo

Lastly they also add some additional blocks to their network that increase performance under certain conditions without significant time loss.

**Strengths:**

Originality:
- Paper combines multiple novel techniques into a single algorithm to achieve a time-save of several orders of magnitude during inference.

Quality:
- Network architecture placed first in a public online tournament for Gomoku
- Paper claims to have created a network architecture that obtains high scores at a significantly faster speed than contemporary algorithms under a fixed-time game of Gomoku. Results from tables and graphs aptly support and verify this claim.
- The paper provides multiple descriptions of all novel algorithms used, as well as descriptions of other algorithms used to compare to it.
- demonstrated a weakness of their method when comparing the loss values of the supervised learning compared to ResNet.

Clarity:
- Paper provides multiple diagrams to effectively communicate the nuances of their algorithm.
- Graphs are general well designed and demonstrate

Significance:
- Incremental update of feature maps, section 3.4, is an interesting concept that could see application beyond the game of Gomoku.

**Weaknesses:**

Impact (Originality and how this paper adds to scientific knowledge):
- The main weakness I see in this paper is its low impact and contribution to the field.
- The paper, at its core, takes policy and value estimates learned by another algorithm (Katagomo), and condenses these estimates into a network architecture that can apply inference significantly faster than ResNet and Katagomo itself.
- The compacting nature of this paper is not necessarily a detriment in and of itself, however the methods used in transforming the slower Katagomo method into the faster Mixnet method should then be applicable beyond the game of Gomoku.
- This is not the case, Two of the methods used to effectively speed up the Katagomo algorithm, in the form of Mixnet, are very specific to Gomoku itself.
- Method 1: decomposing the board as local patterns. The specific line patterns used in this decomposition are very helpful for Gomoku (connecting 5 pieces in a row), but have little application outside of Gomoku.
- Method 2: Distilling pattern based code-book. The fact that we need to iterate over all N board patterns means this method will not scale well to tasks as their state space grows larger and larger. This method also consumes a large amount of memory (scaling by state space). The original mapping network itself is also not very novel, it is just a specialized convolutional filter designed for Gomoku.
- It would be a much more impactful method if these two core methods were able to be used or extended to other environments of varying sizes. Particularly environments with much larger state spaces.

Soundness:
- This paper does have a section addressing the higher loss values of Mixnet compared to ResNet, as I had pointed out in the strength's sections.
- However there is a crucial question I could not find addressed in the paper: If we were to remove the time restriction on moves, how does the ELO difference between Mixnet and its contemporaries change?
- I recognize that the Gomoku tournament itself had time-restricted moves, but again from a generalizability standpoint it is important for people who want to build off of this work to recognize what performance they are giving up (if any) to expedite the inference process.
- This way we get a more wholistic picture where Mixnet is significantly faster, but we do lose X amount of performance if we unbound the time. Or if we lose no performance at all, then this would make the paper significantly stronger: to be able to provide evidence that not only is Mixnet significantly faster, but we also don't lose any performance compared to unbound contemporary algorithms.

**Questions:**

1. Is there a chance to provide an anonymous code submission for your work? I could not find any in the description of the paper.

2. How does Mixnet fair against time-unbound Katagomo and ResNet?

3. In section 3.3 could you explain the computation of the variable "N". It is unclear to me why we are only considering border in this computation, and how the computation makes sense given a 15x15 board (why does i,j = [0,5])?

4. What is the justification for using ResNet as a point of comparison? Was this the previous SOTA?

---

### Official Review · Reviewer_MZL5 · 2024-11-04

**Soundness:** 3
**Presentation:** 2
**Contribution:** 2
**Rating:** 3
**Confidence:** 5

**Summary:**

Rapfi distilled a compact neural network with a pattern-based codebook from CNNs.
The compact NN can convert local line-shaped patterns into features.
With such a faster NN and a codebook for lookup, Rapfi can conduct a much deeper search.
The paper also provided a new way to update the results when only a few stones on the board changed.
In addition, the paper has provided other methods to improve efficiency.
As a result, Rapfi became the GomoCup 2024 champion.

**Strengths:**

1 The agent ranked first in GomoCup2024.
2 The idea of minimized the computation when only a few stones are changed makes sense.
3 The mixnet can be at least 50 times faster than CNN networks.

**Weaknesses:**

1 Need more environments. If the method is general enough and can distill NN correctly, it should be able to be used in other black-and-white-stone board games.
2 It is no surprise that searching deeply can be better in simple games like Gomoku. Moreover, NN is designed for GPUs. Without the comparison of the GPU version doesn't make sense. A small ResNet does not need a high-end GPU.
3 The paper didn't mention any other pattern methods of other board games. They may distill better than the paper's method. For example, there are many distill methods in the game of Go. However, since Go is too complicated, in the end, people give up those methods and only use CNNs.

**Questions:**

1 It is well known that searching deep and searching better on critical nodes are both important.
Have you thought of combining the Resnet and the mixnet in the search [1]?
[1] Multiple Policy Value Monte Carlo Tree Search.
2 A way to test your Mixnet is to use it to replace the Resnet output separately.
3 what is 1p means in table 2?

---

### Note · Authors · 2024-11-14

I have read and agree with the venue's withdrawal policy on behalf of myself and my co-authors.